# Characterization of a Bioactive Peptide T14 in the Human and Rodent Substantia Nigra: Implications for Neurodegenerative Disease

**DOI:** 10.3390/ijms232113119

**Published:** 2022-10-28

**Authors:** Susan Adele Greenfield, Giovanni Ferrati, Clive W. Coen, Auguste Vadisiute, Zoltan Molnár, Sara Garcia-Rates, Sally Frautschy, Gregory M. Cole

**Affiliations:** 1Neuro-Bio Ltd., Building F5, Culham Science Centre, Abingdon OX14 3DB, UK; 2Faculty of Life Sciences & Medicine, King’s College London, London SE1 1UL, UK; 3Department Physiology, Anatomy and Genetics, University of Oxford, Sherrington Building, Parks Road, Oxford OX1 3PT, UK; 4Department of Neurology & Medicine, David Geffen School of Medicine at UCLA and Veterans Affairs Healthcare System, Los Angeles, CA 90095, USA

**Keywords:** acetylcholinesterase, T14, NBP14, dopamine, substantia nigra, Alzheimer’s disease, Parkinson’s disease, voltage sensitive dye imaging

## Abstract

The substantia nigra is generally considered to show significant cell loss not only in Parkinson’s but also in Alzheimer’s disease, conditions that share several neuropathological traits. An interesting feature of this nucleus is that the pars compacta dopaminergic neurons contain acetylcholinesterase (AChE). Independent of its enzymatic role, this protein is released from pars reticulata dendrites, with effects that have been observed in vitro, ex vivo and in vivo. The part of the molecule responsible for these actions has been identified as a 14-mer peptide, T14, cleaved from the AChE C-terminus and acting at an allosteric site on alpha-7 nicotinic receptors, with consequences implicated in neurodegeneration. Here, we show that free T14 is co-localized with tyrosine hydroxylase in rodent pars compacta neurons. In brains with Alzheimer’s pathology, the T14 immunoreactivity in these neurons increases in density as their number decreases with the progression of the disease. To explore the functional implications of raised T14 levels in the substantia nigra, the effect of exogenous peptide on electrically evoked neuronal activation was tested in rat brain slices using optical imaging with a voltage-sensitive dye (Di-4-ANEPPS). A significant reduction in the activation response was observed; this was blocked by the cyclized variant of T14, NBP14. In contrast, no such effect of the peptide was seen in the striatum, a region lacking the T14 target, alpha-7 receptors. These findings add to the accumulating evidence that T14 is a key signaling molecule in neurodegenerative disorders and that its antagonist NBP14 has therapeutic potential.

## 1. Introduction

Despite its established enzymatic role in cholinergic transmission, acetylcholinesterase (AChE) has been acknowledged for some 50 years to be present in a variety of neurons and non-neuronal cells in the absence of its eponymous substrate acetylcholine (ACh) [1]. Although its catalytic action is achieved as a membrane bound enzyme, AChE has been shown to be released from neurons in a potassium-evoked, calcium-dependent fashion in brain areas associated with neurodegenerative diseases. One such site is the substantia nigra (SN) [2,3,4,5,6,7,8,9,10], where its basal and evoked release has been visualized in real-time [11] and monitored on-line in freely moving animals [12,13].

Further evidence suggests that AChE acts as a signaling molecule independent of cholinergic transmission; thus, when its primary catalytic site is irreversibly blocked with the organophosphorus inhibitor Soman [6], it still elicits electrophysiological [14,15,16,17,18,19], biochemical/pharmacological [20,21,22], and behavioral [23,24,25,26] effects. Conversely, butyrylcholinesterase, which also hydrolyses ACh, has no such actions [26,27,28].

The salient part of the AChE molecule underlying these non-enzymatic effects has been identified as T14, a 14-mer peptide derived from the C-terminus of AChE that acts independently of the parent molecule and shares a partial sequence homology with amyloid [29]. We have previously shown that T14 acts at an allosteric site on alpha-7 nicotinic receptors to enhance calcium entry [30] and upregulate the receptor [31], with the potential to trigger a toxic cascade within the cell [32].

The SN is of interest not only due to its centrality in Parkinson’s disease (PD), but also because of its involvement in Alzheimer’s disease (AD), two conditions that share several neuropathological features [29,33,34]. We have previously reported elevated T14 in the AD midbrain, where its content is doubled in comparison with age-matched controls without AD [35]. Given the involvement of the SN in the non-canonical actions of AChE [27], for which T14 is the active component [36], the present study was designed to establish basic features of the non-pathological and pathological distribution of T14 and its physiological actions as a possible signaling molecule in the SN.

Accordingly, we investigated the presence of T14 in the SN, initially in the normal mouse brain and subsequently as a function of Braak staging in the Alzheimer’s brain. To examine the possibility that T14 acts as a signaling molecule in the SN, we used Voltage Sensitive Dye Imaging (VSDI) in ex vivo rat brain slices, as we have previously reported for large scale neuronal activity in the association and sensory cortices and basal forebrain [37,38]. Optical imaging with voltage-sensitive dyes enables the visualization of extensive yet highly transient coalitions of neurons (assemblies) operating throughout the brain on a millisecond time scale [39,40]. In these studies, a 30-mer peptide T30 [31] was used; this contains T14 as its active component, plus the N-terminus lysine and the functionally inert C-terminus 15-mer T15 [35,37]. Because T30 is more stable in solution than T14, it is more practical in experimental applications [31]. We have now investigated the effects of T30 on neuronal activity in the SN. Given that a cyclic form of the T14 peptide, NBP14, has been shown to block the actions of T14 in vitro [35], ex vivo [41] and in vivo [42], it was important to examine whether this antagonist may be of benefit under conditions in which excess T14 is applied to the SN in rat brain slices. Since T14 binds to alpha-7 receptors, we also tested its bioactivity in the striatum, a site that lacks this cholinergic receptor subtype [43] in contrast to the SN, where it is plentiful [32].

The SN shows significant neuronal loss in Parkinson’s and Alzheimer’s disease [29,33,34]. Given the evidence that T14 acts as a signaling molecule in cascades that drive neurodegeneration [42], the present research was undertaken to investigate this peptide within the SN and explore the therapeutic potential of its antagonist NBP14.

## 2. Results

### 2.1. Mouse Brain: T14 Is Co-Localized with Tyrosine Hydroxylase in Pars Compacta Neurons

The T14 antibody used in this study does not recognize the full AChE molecule; it requires an exposed COOH-terminal lysine (see Section 4). Accordingly, it detects T14 free from the parent molecule, AChE, which we have previouly identified in the SN intraneuronally and extracellularly at light- and electron-microscopic levels [44]. Using double-label immunofluorescence, with laser-scanning confocal microscopy, we now show the respective distributions within the SN of T14 (Figure 1a,d,g,j) and the primary enzyme in dopamine biosynthesis, tyrosine hydroxylase (Figure 1b,e,h,k), in mice at postnatal day 21. The results indicate that T14 and tyrosine hydroxylase are co-localized in the pars compacta (Figure 1c,f,i,l).

### 2.2. Human Brain: T14 Immunoreactivity in the Human SN Reflects AD Severity

Coronal sections of human brain containing the SN from Braak II (early) and Braak V (late) stages were assessed for T14 immunoreactivity. T14-immunoreactive neurons were detected at this site at both stages (Figure 2a,b); there was a significant decrease in their number from 84 ± 4 cells/mm^2^ at Braak II to 63 ± 2 cells/mm^2^ at Braak V (Figure 2c; *p* = 0.007, One-Way ANOVA F value (1, 137) = 7.595), confirming cell loss at this site. Although the number of the neurons decreased with AD severity (Figure 2c), there was a significant increase in the ratio of the neurons exhibiting dense versus light T14-immunoreactivity (Figure 2e), indicating an intraneuronal accumulation of T14. The specificity of the T14 immunoreactivity in this preparation was tested. When the primary antibody was immunoneutralized with T14, no immunoreactivity was observed; under those circumstances, the neuromelanin-containing pars compacta dopaminergic neurons were unmasked and became visible (Figure 2d).

### 2.3. Rat Brain: Characterization Ex Vivo of T14/T30 Action

Voltage sensitive dye imaging has been used to to evaluate the spatio-temporal dynamics of individual highly transient, large-scale, neuronal assemblies in brain slices following electrical stimulation [37,40,45]. Here, the technique was used to investigate the potential actions of the T30 peptide on neuronal assemblies in the SN or striatum, sites at which alpha-7 nicotinic receptors are, respectively, present or absent. Each experimental session comprised three 20-min perfusion epochs. The first provided control data in the presence of artificial cerebrospinal fluid (ACSF) alone. To establish the viability of the slices throughout the experiment and the absence of any significant photobleaching, the second and third epochs involved the presence of ACSF alone (Figure 3a,c). Otherwise, T30 was present in the perfusate throughout the second and third epochs, with or without NBP14, the antagonist that blocks its binding to alpha-7 receptors, in the third epoch.

Optical imaging of the SN in rat brain slices showed that the presence of T30 attenuated electrically evoked neuronal activity when compared with recordings made in the presence of ACSF alone (Figure 3a and Section 4 Methods). When NBP14 was present, the attenuating action of T30 on evoked neuronal activity in the SN was blocked (Figure 3b). In contrast, the electrical stimulation in the striatum showed a markedly higher amplitude response, with greater spatial diffusion, and the presence of T30 had no effect on that response (Figure 3c). These results demonstrate the site-dependent responsiveness to the peptide and provide further evidence for its dependency on alpha-7 nicotinic receptors.

## 3. Discussion

For over forty years PD and AD have been viewed as disorders with overlapping processes of neurodegeneration, the primary cell loss occurring within a hub of neuronal nuclei characterized by isodendritic cytoarchitecture [46] and sharing embryologic provenance: the basal rather than alar plate [47]. Evidence has accumulated [48,49,50,51,52,53,54,55] indicating that these isodendritic nuclei are the first to exhibit pathology in AD, in advance of the hippocampus and cerebral cortex [56]. We now show that T14, the 14-mer peptide cleaved from the AChE C-terminus, may be a significant feature in the anatomy, physiology and pathology of one of these nuclei, the SN.

We have previously reported the presence of AChE in the pars compacta, pars reticulata and extracellular space of the SN at light- and electron-microscopic levels [44]. The present immunohistochemical studies used a polyclonal antibody against T14 that requires an exposed COOH-terminal lysine, thereby precluding the detection of AChE, the parent molecule, or amyloid, with which T14 has a partial sequence homology [35,42]. Accordingly, we have identified free T14 in pars compacta neurons of the mouse SN; these were shown to be dopamine neurons by the colocalization of tyrosine hydroxylase. A comparable profile was seen for T14 in the adult human SN. Its location in the dopamine neurons was indicated by the masking of the neuromelanin deposits with the T14 immunoreactivity; those deposits became visible when the antibody had been immunoneutralized. The discovery of T14 immunoreactivity in SN neurons is consistent with our earlier research on the human midbrain in which Western blot analysis showed the concentration of the peptide to be approximately twice as high in AD samples compared with age-matched controls without the disease [35]. The present study employed immunohistochemistry to focus on the SN and found an increasingly dense T14-immunoreactive signal as AD progresses from Braak Stage II to Braak Stage V. This observation suggests an involvement of T14 at this site.

T30 is a 30-mer peptide from the C-terminus of AChE; it includes T14 as the active component and is more stable than the smaller peptide in solution [31]. T14 and T30 act via alpha-7 receptors [30,31], enhancing calcium entry through an allosteric site [30] and triggering the accumulation of phosphorylated Tau and amyloid beta [35,42]. Previous studies using various experimental preparations have demonstrated that the binding of T14 or T30 to alpha-7 receptors can be blocked by NBP14, a cyclized variant that displaces the peptide from its receptor [33,35,37]. The dopamine neurons in the SN have been shown to possess at least two subtypes of postsynaptic nicotinic receptors, alpha-7 and alpha-4-beta-2 [57]; the presence of the alpha-7 subtype in those neurons was confirmed when single-cell RT-PCR, following electrophysiological characterization, identified the mRNAs for TH and the receptor [58].We now report that the effects of T30 on electrically evoked activity within the SN were blocked by NBP14, as previously observed in the association and sensory cortices and the basal forebrain [37,38,59]. In contrast, under comparable experimental conditions, T30 had no effect on evoked activity in the striatum, a site rich in a range of nicotinic receptors, but lacking the alpha-7 subtype [39], the target for T30 [31].

The present studies on rodent brain slices show that treatment with T30 at 2µM reduces the magnitude of the electrically evoked activity in the SN. Comparable research on the basal forebrain [59] also found an attenuating effect with this concentration in contrast to the enhanced activity seen with 1 µM. Our previous studies on oocytes suggest that the peptide, through activation of an allosteric site on alpha-7 receptors, has dose-dependent effects on calcium influx such that excessive intracellular calcium induced by the higher concentration inactivates the channels [30,60] leading to the attenuation of the evoked activity seen here. The experimental paradigm used in the present VSDI studies is acute; under pathological conditions of chronic exposure to elevated levels of the peptide, the excessive intracellular calcium may be expected to precipitate the pathological cascades described in Greenfield et al. [42]. We have previously reported that chronic exposure to T14 has a toxic effect on cultured hippocampal neurons that can be reversed by a CaM kinase II inhibitor, indicating that longer term pathological actions may be mediated via mitochondrial dysfunction [32].

Basal plate-derived neurons, including those in the SN, have a range of distinguishing features such as spontaneous activity in vitro, diffuse aminergic projections and, of particular note in the present context, a sensitivity to trophic agents that continues into adulthood [61]. It has been postulated that this persisting developmental mechanism accounts for the primary vulnerability of these cell groups in AD [29]. Calcium influx is an extracellular trigger for cell growth [62], but intracellular calcium tolerance can be markedly reduced with age [63]. Hence, activation of the T14 system in the adult brain, for example, in response to damage, may set in train a response that would be appropriate in a developmental context but deleterious in the mature brain, thereby initiating a cycle of neuronal death [42].

In conclusion, the current study adds to the evidence that T14, the AChE-derived peptide that acts on alpha-7 receptors, may be a significant player in neurodegeneration. The peptide is not only prominent and physiologically active in the rodent SN, but also significantly elevated at that site in AD. If the SN is susceptible to neuronal degeneration through the actions of T14, its antagonist, NBP14, may have therapeutic potential in the early stages of both AD and PD.

## 4. Materials and Methods

### 4.1. Mouse Brains

Animal experiments were approved by a local ethical review committee and conducted in accordance with the UK Animals (Scientific Procedures) Act, 1986 (ASPA), under valid personal licences and project licence PPL P1E785A22.

C57BL/6J mouse brains for immunohistochemistry at postnatal day 21 were perfusion-fixed with 4% formaldehyde (F8775; Sigma-Aldrich, Gillingham, UK) in 0.1 M phosphate-buffered saline (PBS). Hemispheres were sectioned coronally at 50 μm on a vibrating microtome (VT1000S; Leica Biosystems, Nussloch, Germany). Free-floating sections were blocked with 5% donkey serum and 0.1% Triton-X100 for 2 h before incubation with primary antibodies: sheep anti-TH (1:250, ab113; Abcam, Cambridge Biomedical Campus, UK) and rabbit anti-T14 for 48 h at 4 °C. The T14 antibody was raised against the 14-mer peptide, the active part from the C-terminus of AChE. It has been previously characterized for Western blots and immunohistochemistry [35,42] and fails to recognize the following: amyloid, the full AChE molecule, T30, T15 (the inert 15-mer sequence at the C-terminus of T30) or T14 without an exposed COOH-terminal lysine. These results indicate that it binds to the VHWK-COOH terminal region in T14 and that binding is end-specific for the free COOH. Sections were washed in 0.1 M PBS before incubation with secondary antibodies: donkey anti-rabbit IgG AlexaFluor488 (1:500; A21206, Invitrogen, Inchinnan, UK) and donkey anti-sheep biotinylated (1:200; ab6899, Abcam, Cambridge Biomedical Campus, UK) in blocking solution at room temperature (RT) for 2 h. Sections were additionally incubated with streptavidin-Cy5 (1:500, SA1011; Molecular Probes, Leiden, The Netherlands). All the sections were counterstained with 4,6-diamidine-2-phenylindole dihydrochloride (DAPI; 1:1000; Invitrogen, Inchinnan, UK) and mounted with Fluorsave (345789, Millipore, Darmstadt, Germany). Immunolabeled sections were imaged with a laser-scanning confocal microscope (Zeiss LSM710, Berlin, Germany) using 5×, 10×, 20× objectives and 0.6 or 1 optical zoom at 0.42 μm pixel size, and frame size 1024 × 1024.

### 4.2. Human Brains

Human midbrain blocks were provided by Dr Wayne Poon, Director of UCI MIND Tissue Repository Gillespie Neuroscience Research Facility at University of California, Irvine and the National Brain Bank (https://neurobiobank.nih.gov/about/, accessed on 7 October 2019) from Braak II and Braak V, ages, respectively, 85.8 ± 1.32 and 85.6 ± 0.93 years, male and female. The human tissue studies were performed according to Greater LA VA IACUC Project 1615970-4.

Human tissue was fixed in 10% formalin (Fisher Scientific, Waltham, MA, USA, cat #245-685) followed by dehydration and embedding in paraffin; 6 µm sections were cut and cleared with Citri Solv (Decon Labs Inc, King of Prussia, PA, USA); 3% hydrogen peroxide (30 min) was used to quench endogenous peroxidase activity followed by antigen retrieval in Ph9 buffer (Sigma-Aldrich, Burlington, MA, USA) containing 2 mM EDTA (121 °C; 10 min). Sections were pre-treated with tris buffered saline containing 0.5% tween 20 (TBS-T) to permeabilize. Primary stock affinity purified polyclonal rabbit anti-T14 (1 mg/mL; Genosphere Biotechnologies, Paris, France) was used at 1:200 (1 h). Vector kits (Vector Lab, Burlingame, CA, USA) were used with secondary biotinylated goat anti rabbit IgG (Vector Lab, Burlingame, CA, USA, Cat #BA-1000) at 1:500 (60 min). HRP substrate was applied and incubated (75 min) before washing with TBS-T (2 × 5 min). Vector VIP Substrate was applied (15 min; Cat #SL -4600, Vector Lab, Burlingame, CA, USA. There was an absence of immunoreactivity when the primary antibody was omitted. For the immunoneutralization experiments, 1 µg/mL of T14 in 1:200 dilution of anti-T14 was used.

Image analysis used ImageJ (https://imagej.nih.gov/ij/download.html, accessed on 16 August 2022; National Institutes of Health, Bethesda, MD, USA). The macro included several automated steps as follows: RGB image translated into an RGB stack with red slice to analyze purple chromogen excluding melanin. The macro calibrated the image, translating intensity of staining to relative optical density (OD) and pixels to micrometers; minor adjustments made to reduce signal to noise such as despeckle, smooth or enhance contrast, were decided prior to analysis of sections according to the staining and applied automatically to all images acquired. The same density thresholding was chosen, and particles were analyzed according to small, then large particle size differences. The final steps in this macro evaluated dark vs. light particles. A blinded experimenter conducted the analysis on 9 sections per subject: relative OD, size, cells per area of dark and light. Cell density (cell count/µm^2^) required calculation of the region of interest (ROI) chosen for analysis. Since only percent area and total area stained are provided by ImageJ (National Institutes of Health, Bethesda, MD, USA), ROI is calculated by (100 × total area stained × percent area^−1^, variables built-in to ImageJ (National Institutes of Health, Bethesda, MD, USA). Count of areas examined was then divided by ROI × 1,000,000.

### 4.3. Ex Vivo Rat Brain Slice Preparation

Male Wistar rats (Charles River, Harlow, UK) were used for the optical imaging experiments at postnatal day 21, a developmental stage within the range previously shown to exhibit the conspicuous assemblies that indicate large-scale neuronal cohesion [37,45]. All animal procedures were carried out according to the Home Office UK regulations, in compliance with the requirements of the UK Animals (Scientific Procedures) Act 1986.

Brain slice preparation was carried out as follows: consecutive 400 μm thick coronal rat brain slices containing the SN (coordinates −4.80 to −6.20mm from Bregma [64]) or striatum (1.20 to −0.20 mm from Bregma) were cut with a Leica VT1000S vibrating microtome in oxygenated (95% O_2_, 5% CO_2_) ice-cold cutting solution containing the following (in mmol): 120 NaCl, 5 KCl, 20 NaHCO_3_, 2.4 CaCl_2_, 2 MgSO_4_, 1.2 KH_2_PO_4_, 10 glucose, 6.7 HEPES salt and 3.3 HEPES acid; pH: 7.1. Each slice was divided along the midline into hemisections; these were transferred to a bubbler pot with artificial cerebrospinal fluid (ACSF) containing (in mmol): 124 NaCl, 3.7 KCl, 26 NaHCO_3_, 2 CaCl_2_, 1.3 MgSO_4_, 1.3 KH_2_PO_4_ and 10 glucose; pH: 7.1. They were incubated at 34 °C for 30 min. The hemisections were then kept at RT (22 ± 1.5 °C) for 30 min in oxygenated ACSF before VSD staining.

### 4.4. Voltage Sensitive Dye Imaging

Each VSDI experiment was performed as previously described [65]. Briefly, slices were incubated in a dark, high humidity chamber filled with an oxygenated voltage-sensitive dye solution containing Di-4-ANEPPS (4%; dissolved in ACSF, foetal bovine serum 48%, DMSO 3.5% and Cremophor EL 0.4%) for 20 min at RT. Sections were then transferred to a bubbler pot with oxygenated ACSF at RT and kept for 45 min to wash off excess dye and favour the recovery phase before the VSDI recordings. The dye was chosen because it is characterized by minimal pharmacological side effects or phototoxicity and a high signal-to-noise ratio [65,66].

After incubation with the dye, hemisections were placed in the recording bath, continuously perfused with oxygenated ACSF and warmed to 30 ± 1 °C with a temperature control system (TC-202 A, Digitimer Research Instruments, Hertfordshire, UK). Slices were kept in position with a home-made plastic grid before placing a concentric bipolar platinum/iridium (Pt-Ir) stimulating electrode (outer pole diameter 200 μm, inner pole diameter 25 μm; FHC, Bowdoin, ME, USA) in either the pars compacta of the SN or the striatum. VSDI neuronal network responses in the ROI were evoked with the delivery of an electrical stimulation consisting of a 100 µs pulse of 30 volts, with an inter-stimulus interval (ISI) of 28 s. VSDI data were collected and analysed as previously described [38]. Excitation light inducing fluorescence was provided by a halogen lamp source (Osram xenophot 64634 HLX EFR Display/Optic lamp) and filtered to emit green light (530 ± 10 nm) using a MHF-G150LR fiber optic light source (Moritex Corporation, Asaka, Saitama, Japan) coupled to MiCAM 02-HR ultra-fast imaging system. The emitted fluorescence was filtered through a >590 nm high-pass filter. 16-bit images were recorded with 3.7 ms resolution (detector array of 184 × 124 pixels) using a digital camera (Brain Vision MiCAM 02-HR, Tokyo, Japan) controlled by Ultima 2004/08 imaging software (Brain Vision, Tokyo, Japan) coupled to Spike 2 V6.0 (CED Ltd., Cambridge, UK), which was employed to trigger stimulations every 28 s. Each VSDI acquisition was recorded for 400 ms around the stimulus onset, including a 100 ms pre-stimulus baseline providing the background activity, and a 300 ms post-stimulus period. To quantify VSDI signals, the fractional change in fluorescence over baseline (Σ∆F/F_0_) between 0 and 300 ms after electrical stimulation was calculated, where F_0_ is the average absolute fluorescence for the 100 ms preceding stimulation onset. The software averaged the approximate 20 bins per recording condition and provided an averaged image per slice per recording condition. The space-time maps shown here give the overall average of each recording condition for the total number of slices.

Each experimental session comprised three 20-min perfusion epochs, the first 10 min for acclimatization to the recording bath and the second 10 min for the recording. The first epoch provided control data in the presence of ACSF alone; this was followed by a second and a third epoch, which differed according to the experimental design. For the studies undertaken to establish the viability of the slices throughout the experiment and the absence of any significant photobleaching (Figure 3a,c), the second and third epochs involved the presence of ACSF alone. For the T30 experiments, 2 µM T30 was present in the perfusate throughout the second and third epochs; for the NBP14 experiments, 2 µM T30 was present in the second and third epochs, with the addition of 4 µM NBP14 in the third epoch.

### 4.5. Data Analysis

Optical imaging data from all stimulations were averaged into a single file for each experimental condition and analysed using a VSDI data analysis toolbox [67] implemented in MatLab (The Mathworks Inc, Natick, MA, USA). Briefly, this toolbox provides the possibility to select a fixed ROI geometry which is applied to every slice to extract relevant data from an identical ROI across all slices and experimental conditions. The ROI was post hoc overlaid onto the imaged slice to include the entirety of evoked VSDI responses of the areas analysed in this study. VSDI data from the selected ROIs (SN and the striatum) were then plotted as the magnitude of activity over space and time (‘space-time’ maps), and as histograms of the average of summed long-latency fluorescence fractional change over baseline between 0 and 300 ms after stimulation delivery (Σ∆F/F_0_). The toolbox uses a custom color map to display spatiotemporal activity, with warm and cool colors representing, respectively, depolarization and hyperpolarization. Statistical analysis was performed using GraphPad Prism 6 (v6.05; GraphPad Software Inc., San Diego, CA, USA). Since all the data were tested for normality, only parametric tests (ANOVA followed by Fisher LSD post hoc tests) were used.

### 4.6. Statistical Significance

For all statistical tests, *p* < 0.05 was considered significant; data are expressed as mean ± SEM (standard error of the mean).

## Figures and Tables

**Figure 1 ijms-23-13119-f001:**
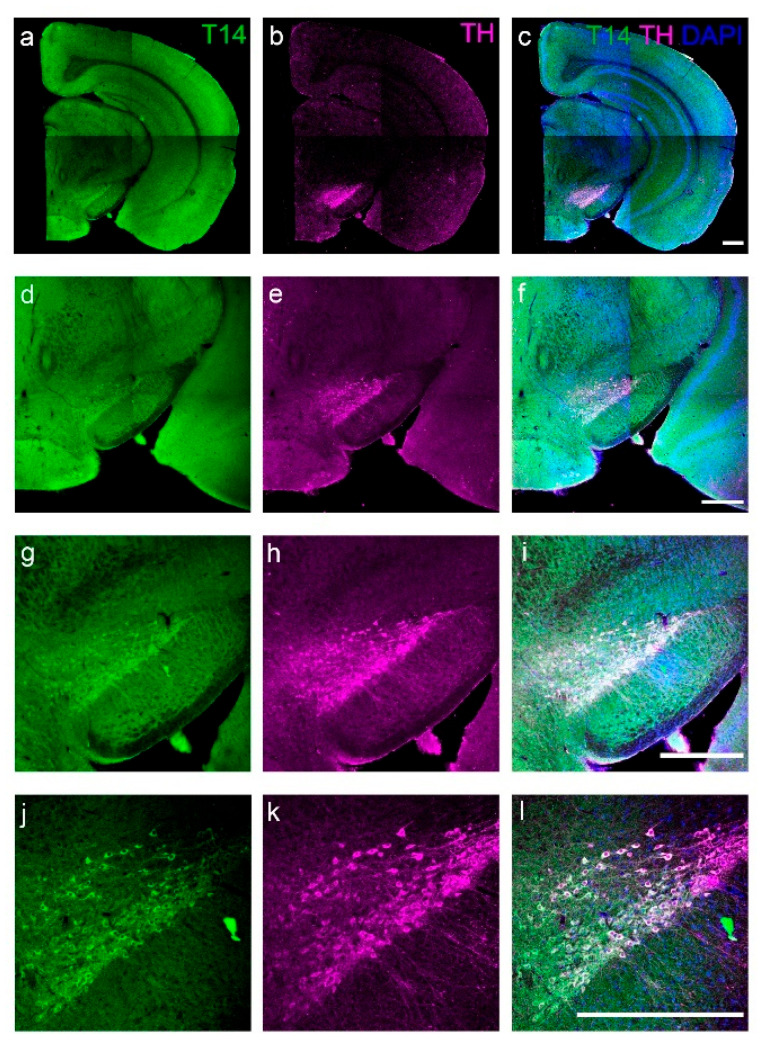
Photomicrographs of a representative coronal section of a mouse brain on postnatal day 21 at the rostro-caudal level of the substantia nigra showing immunofluorescence for T14 (appears green in (**a**,**d**,**g**,**j**)) and tyrosine hydroxylase (appears magenta in (**b**,**e**,**h**,**k**)), the primary enzyme in dopamine biosynthesis. Images at increasing magnification (**c**,**f**,**I**,**l**) demonstrate the co-localization of these factors within the cell bodies of the pars compacta of the substantia nigra. Cell nuclei are shown by DAPI fluorescence (blue). Scale bars: 500 µm.

**Figure 2 ijms-23-13119-f002:**
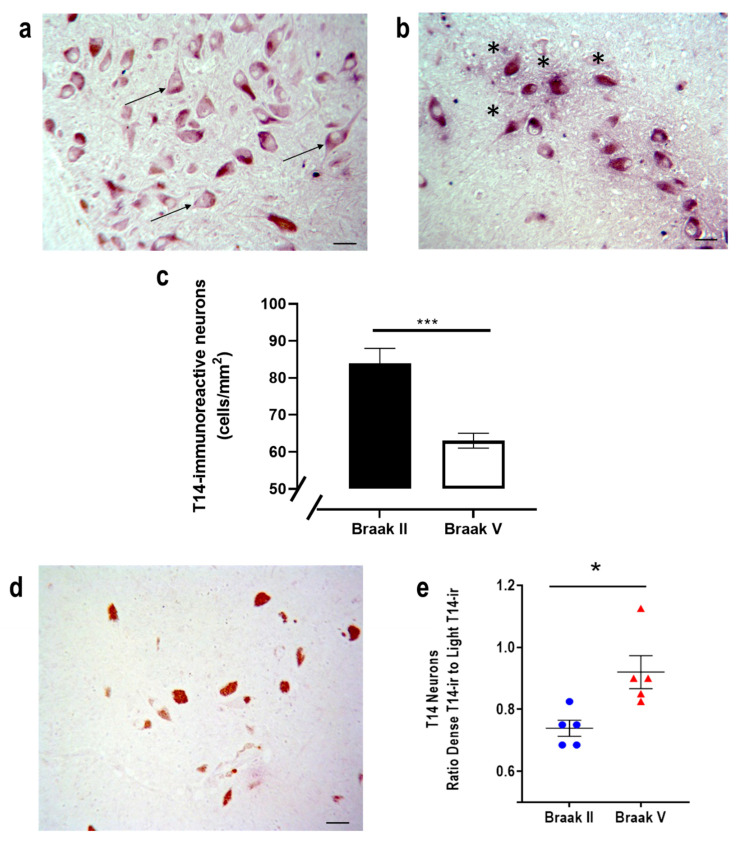
(**a**,**b**,**d**) Representative photomicrographs of human substantia nigra with T14 immunoreactivity (scale bars: 50 µm). T14 immunoreactivity at (**a**) Braak II (arrows: examples of lightly immunoreactive neurons) and (**b**) Braak V (asterisks: examples of densely immunoreactive neurons). (**c**) Quantification of cells in Braak II and V (n = 5 subjects in each group, *** *p* = 0.001, unpaired *t*-test). (**d**) Absence of T14 immunoreactivity following immunoneutralization of the primary antibody with T14; the neuromelanin is now visible. (**e**) Ratio of dense T14-immunoreactive to light T14-immunoreactive neurons in the substantia nigra at Braak II and Braak V (n = 5 subjects in each group, * *p* < 0.05, unpaired *t*-test).

**Figure 3 ijms-23-13119-f003:**
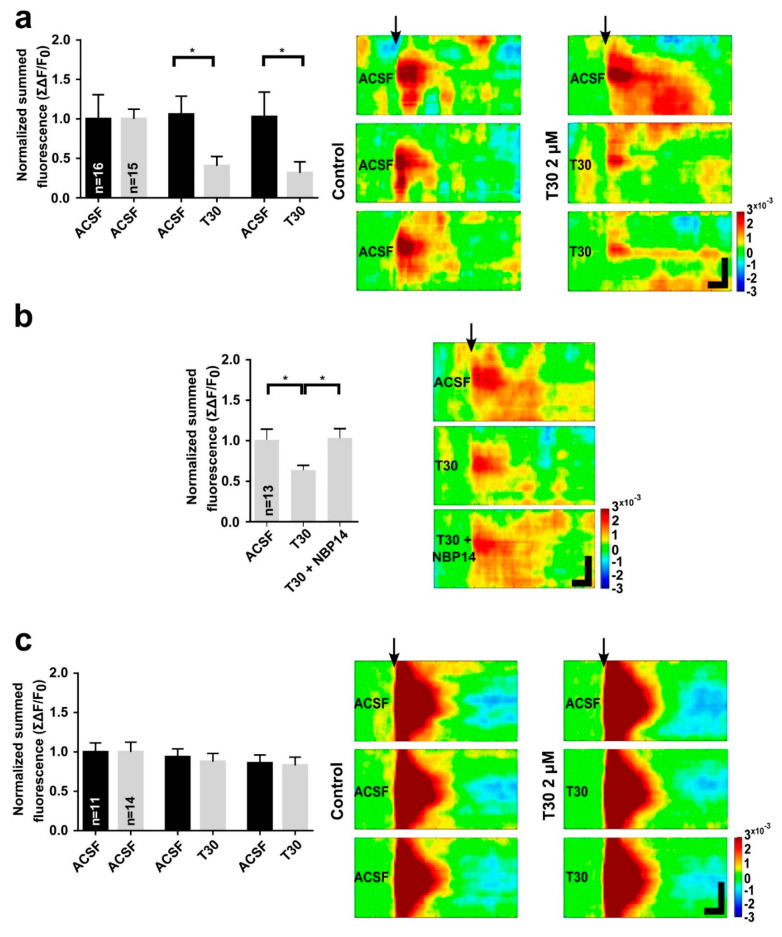
Optical imaging of the substantia nigra (**a**,**b**) or striatum (**c**) with a voltage sensitive dye (Di−4−ANEPPS) in rat brain slices at postnatal day 21. The response to electrical stimulation in the three sequential epochs (see Section 4) is shown as summed data in bar graphs (sequence from left to right) and as space-time maps (sequence from top to bottom). (**a**) The presence of T30 (2 µM) caused a significant attenuation of the electrically evoked response (grey bars) compared with control recordings (black bars) made in the presence of artificial cerebrospinal fluid (ACSF) alone (One-way ANOVA, F (1.503, 21.04) = 7.74, *p* = 0.0055; post hoc Fisher’s LSD test, *p* < 0.05). (**b**) Including NBP14 (4 µM) with T30 (2 µM) prevented attenuation of the evoked response (One-way ANOVA F (1.819, 21.82) = 3.555, *p* = 0.0499; post hoc Fisher’s LSD test, *p* < 0.05). (**c**) When the striatum was exposed to T30 (2 µM), the electrically evoked response was unaffected. The black arrows mark the onset of the stimulus in the space-time maps, for which the horizontal and vertical scale bars represent 50 ms and 1 mm, respectively. Data are the mean ± SEM. n = (hemislices, rats). For (**a**): n = (16 or 15, 4). For (**b**): n = (13, 4), For (**c**): n = (11 or 14, 3). * *p* < 0.05.

## Data Availability

Not applicable.

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
