# Peer review of "Characterization of a Bioactive Peptide T14 in the Human and Rodent Substantia Nigra: Implications for Neurodegenerative Disease"

_ijms, 2022, doi:10.3390/ijms232113119_

Round 1
Reviewer 1 Report
The work submitted is complementary to one recently published by the authors (A novel process driving Alzheimer's disease validated in a mouse model: Therapeutic potential) and adds to the evidence that T14 may be a significant player in neurodegeneration. Herein, the authors have identified free T14 in dopamine neurons of rodent substancia nigra, while T14 distribution in human SN is comparable. The cyclized variant of T14, namely NBP14 acts as an antagonist with therapeutic potential.
In section “2.2. Human brain: T14 immunoreactivity in the human SN reflects AD severity”, the authors should indicate the correct figure within text (Figure 2 instead of Figure 1).
Author Response
In section “2.2. Human brain: T14 immunoreactivity in the human SN reflects AD severity”, the authors should indicate the correct figure within text (Figure 2 instead of Figure 1).
RESPONSE: We regret this oversight and have made the necessary corrections (lines 98, 99 & 107).
Reviewer 2 Report
In this manuscript, the authors investigated the spatial expression pattern of AchE derived T14 peptide in mouse midbrains and also its pathological and clinical relevance in different Braak stages of human AD brains. Finally the functional role of T30, stable version of endogenous T14 peptide was examined in the rat midbrain slices to show its ability to suppress electrical activities possibly through action on alpha-7 receptor. Extending from their previous findings on T14 expression AD patients brain, there seems some progress on T14 pathophysiological role in SN brain regions. However, more detailed and additional experiments would be required to conclusively support the pathological role of T14 in nigra dopaminergic neuron. The following comments can be considered for potential revision of this manuscript.
- The last paragraph of the introduction can be revised to end more appropriately. The current introduction sounds incomplete, especially the ending sentence.
- The results are overall too short and lack the details of experiments performed. The actual data was not explained faithfully.
- Figure panels need to be arranged in the order as they are called out in the main text.
- In the Figure 1, it is interesting to see the largely overlapping signals of both TH and T14 in nigral dopaminergic neurons. Although the author stated that the T14 antibody used in this study does not bind to AchE, it would be necessary to show the actual expression pattern of the precursor AchE in this SN region as compared with T14 expression. Without AchE expression data, it is not clear how this T14 peptide end up in this DA neuron intracellular compartment.
- I wonder why brain sections from well established AD mouse models were not used in these experiments to look at potential alteration of T14 expression levels in DA neurons in response to AD related pathology. This complete set of data of both control and AD mouse brains will be nicely in line with the following clinical relevance of T14 expression in AD patients brains of advancing stages.
- How about the number of TH positive dopamine neurons in AD patients' postmortem brains? Is postmortem brain from age-matched control not available for comparison of T14 expression? It would be better to provide average T14 signal intensities in a neuron to support the conclusion in the text that T14 levels in DA neurons are increased.
- Double fluorescence labeling would be more informative to demonstrate alteration of dopaminergic T14 expression in different stages of AD. Although the author stated the different stages of the patient's brain was used, still providing the levels of Ab or tau pathologies in the indicated brain sections would be informative to correlate with the extent of T14 expression.
- I wonder why ANOVA test was applied for two sample comparison in panel C.
- Please clarify what the n number in the figure legends actually means?
- Although it is stated in the text, T14 target alpha-7 receptor expression can be shown for the slices used for electrical response imaging. Although the authors used NBP7 to support specificity of T30 action, it would be also important to show that the degree of suppression of neuronal activities by T30 is dose dependent. By the way, do we expect potentiation of neuronal activity with T30 mediated activation of alpha-7 receptor and the subsequent calcium influx? There is neither explanation nor discussion about how this data can be translated into the pathological condition of AD.
- It is recommended to add discussion about the spatial distribution of AchE, T14, and its receptor alpha 7 in SN. From the data presented, it is not at all clear on which cell type of SN alpha 7 receptors are possibly expressed. Discussion about potential mechanisms of the increase of T14 in AD pathogenesis can be also included. How is the suppression of midbrain neuronal activity by T14 translated to pathogenesis of neuronal degeneration in clinicopathological conditions?
- Animal protocol number needs to be provided.
Author Response
Review Report Form – Reviewer 2
- The last paragraph of the introduction can be revised to end more appropriately. The current introduction sounds incomplete, especially the ending sentence.
RESPONSE: We are grateful for this comment and have improved the text accordingly (lines 80 to 83).
- The results are overall too short and lack the details of experiments performed. The actual data was not explained faithfully.
RESPONSE: This is a very helpful comment. We have now expanded the results section to ensure that our findings are more clearly conveyed (lines 86 to 127). The significance of the voltage sensitive dye imaging (VSDI) technique is now clarified by reference to the classic text in this field in Nature Reviews Neuroscience (VSDI: a new era in functional imaging of cortical dynamics. Grinvald & Hildesheim, Nature Rev Neurosci 2004, 5, 874–885).
- Figure panels need to be arranged in the order as they are called out in the main text.
RESPONSE: We regret this oversight and have made the necessary corrections (lines 98, 99 & 107).
- In the Figure 1, it is interesting to see the largely overlapping signals of both TH and T14 in nigral dopaminergic neurons. Although the author stated that the T14 antibody used in this study does not bind to AchE, it would be necessary to show the actual expression pattern of the precursor AchE in this SN region as compared with T14 expression. Without AchE expression data, it is not clear how this T14 peptide end up in this DA neuron intracellular compartment.
RESPONSE: We are grateful for these comments. For clarification we are now citing (lines 89 & 170) our earlier study (Henderson & Greenfield, 1984) in which we reported the presence of AChE in the pars compacta, pars reticulata and extracellular space of the substantia nigra at light- and electron-microscopic levels.
- I wonder why brain sections from well established AD mouse models were not used in these experiments to look at potential alteration of T14 expression levels in DA neurons in response to AD related pathology. This complete set of data of both control and AD mouse brains will be nicely in line with the following clinical relevance of T14 expression in AD patients brains of advancing stages.
RESPONSE: Earlier this year we published our findings on T14 in a well-established AD mouse model (5XFAD) that develops severe amyloid pathology, cited here as reference 42. That study focused on the hippocampus and identified T14-immunoreactive neurons in the immediate vicinity of plaques. Comparable work on the mouse model was considered beyond the scope of the current study on the substantia nigra.
- How about the number of TH positive dopamine neurons in AD patients' postmortem brains? Is postmortem brain from age-matched control not available for comparison of T14 expression?
RESPONSE: This an interesting point. However, it should be noted that Kazee et al. (Alzheimer Disease and Associated Disorders 1995, 9, 61-67) documented the loss of pigmented neurons in the substantia nigra (presumed to be dopamine neurons) in Alzheimer’s disease (now cited in lines 57 & 81). We believe that the present study may inspire further research on this dopamine system; a very extensive study on postmortem tissues would be required to provide useful data. We consider that to be outside the scope of the present study.
It would be better to provide average T14 signal intensities in a neuron to support the conclusion in the text that T14 levels in DA neurons are increased.
RESPONSE: We appreciate the suggestion that quantifying signal intensity at the individual neuron level would be a useful parameter. However, the stereological problems of quantifying the DAB product neuron by neuron make this unrealistic at this stage within the given timeframe.
- Double fluorescence labeling would be more informative to demonstrate alteration of dopaminergic T14 expression in different stages of AD. Although the author stated the different stages of the patient's brain was used, still providing the levels of Ab or tau pathologies in the indicated brain sections would be informative to correlate with the extent of T14 expression.
RESPONSE: We agree that it will eventually be informative to correlate abundance of T14 and Tau or Aβ. Practical constraints in the present study required the Braak staging to be determined at source: Dr Wayne Poon, Director of UCI MIND Tissue Repository Gillespie Neuroscience Research Facility at University of California, Irvine. As we now indicate in the Discussion (lines 188-189), we have already reported (Garcia-Rates et al., 2016) that T30 promotes amyloid beta production and tau phosphorylation via an allosteric site on the alpha-7 nicotinic receptor and that these actions are blocked by NBP14.
- I wonder why ANOVA test was applied for two sample comparison in panel C.
RESPONSE: We are grateful for this appropriate comment and are now presenting the statistics as an unpaired t-test (line 141).
- Please clarify what the n number in the figure legends actually means?
RESPONSE: We have now clarified this important point (lines 141, 144, 158-159 & 286).
- Although it is stated in the text, T14 target alpha-7 receptor expression can be shown for the slices used for electrical response imaging. Although the authors used NBP7 to support specificity of T30 action, it would be also important to show that the degree of suppression of neuronal activities by T30 is dose dependent. By the way, do we expect potentiation of neuronal activity with T30 mediated activation of alpha-7 receptor and the subsequent calcium influx?
RESPONSE: We now realise that we had omitted to provide the dose-response and calcium influx background to the present studies. These key points have been included (lines 202-215).
There is neither explanation nor discussion about how this data can be translated into the pathological condition of AD.
RESPONSE: We agree that this important point needed further elaboration. This has been addressed (lines 80-83 & 202-215). Further explanation and discussion are provided in the key cited reference: Greenfield et al., 2022.That paper presents in detail the proposed mechanisms of action of the T14 system in signaling cascades that drive neurodegeneration. We could recapitulate the description of those mechanisms in the present paper. However, we believe that an exhaustive review would extend the text inappropriately and that the additions we have made will guide the reader appropriately.
- It is recommended to add discussion about the spatial distribution of AchE, T14, and its receptor alpha 7 in SN. From the data presented, it is not at all clear on which cell type of SN alpha 7 receptors are possibly expressed.
RESPONSE: As mentioned above, we are now citing our earlier study (Henderson & Greenfield, 1984) (lines 89 & 170) in which we reported the presence of AChE in the pars compacta, pars reticulata and extracellular space of the substantia nigra at light- and electron-microscopic levels. Regarding alpha-7 in the SN, we are now citing Matsubayashi et al., 2004a and Matsubayashi et al., 2004b (lines 192-196).
Matsubayashi, H. Amano, T. Seki, T. Sasa, M. Sakai, N. (2004a) Postsynaptic α4β2 and α7 type nicotinic acetylcholine receptors contribute to the local and endogenous acetylcholine-mediated synaptic transmissions in nigral dopaminergic neurons. Brain Res 1005, 1-8.
Matsubayashi, H. Inoue, A. Amano, T. Seki, T. Nakata, Y. Sasa, M. Sakai, N. (2004b) Involvement of alpha7- and alpha4beta2-type postsynaptic nicotinic acetylcholine receptors in nicotine-induced excitation of dopaminergic neurons in the substantia nigra: a patch clamp and single-cell PCR study using acutely dissociated nigral neurons. Brain Res Mol Brain Res 129, 1-7.
Discussion about potential mechanisms of the increase of T14 in AD pathogenesis can be also included.
RESPONSE: As indicated above, this important point has been further addressed on lines 80-83 & 202-215. We now steer the reader to the key 2022 reference in which we present the signaling cascades by which T14 is considered to drive the neuropathologies.
How is the suppression of midbrain neuronal activity by T14 translated to pathogenesis of neuronal degeneration in clinicopathological conditions?
RESPONSE: The evidence that a high concentration of the peptide leads to excessive calcium influx (Bon & Greenfield, 2003; Greenfield et al., 2004) suggests that consequent calcium channel inactivation (see Standen, Nature1981, 293, 158-159) underlies the attenuation of the evoked activity observed here. The experimental paradigm used in the present VSDI studies is acute; under pathological conditions of chronic exposure to elevated levels of the peptide, the excessive intracellular calcium may be expected to induce mitochondrial dysfunction and the production of free radicals (see Montal, Biochim Biophys Acta 1998, 1366, 113-126; Greenfield et al., 2022). Moreover, we know that chronic exposure to T14 has a toxic effect on cultured hippocampal neurons that can be reversed by a CaM kinase II inhibitor, indicating that longer term pathological actions may be mediated via mitochondrial dysfunction (Day and Greenfield, 2003). The text has been revised accordingly (lines 202-215).
- Animal protocol number needs to be provided.
RESPONSE: These details are now included in the text (lines 236 & 262-263).
Round 2
Reviewer 2 Report
My previous concerns were addressed sufficiently by the authors' response and revision.